# Magnetic Properties of Amorphous Ta/CoFeB/MgO/Ta Thin Films on Deformable Substrates with Magnetic Field Angle and Tensile Strain

**DOI:** 10.3390/s23177479

**Published:** 2023-08-28

**Authors:** Ah Hyun Jun, Young Hyun Hwang, Byeongwoo Kang, Seungwon Lee, Jiwon Seok, Jong Seong Lee, Seo Hyun Song, Byeong-Kwon Ju

**Affiliations:** Display and Nanosensor Laboratory, School of Electrical Engineering, Korea University, 145, Anam-ro, Seongbuk-gu, Seoul 02841, Republic of Koreaaksyp@korea.ac.kr (Y.H.H.);

**Keywords:** magnetic sensors, in-plane angle sensors, CoFeB, biaxial, four-fold, deformable, flexible, wearable, stretchable

## Abstract

Recently, the application of cobalt iron boron (CoFeB) thin films in magnetic sensors has been widely studied owing to their high magnetic moment, anisotropy, and stability. However, most of these studies were conducted on rigid silicon substrates. For diverse applications of magnetic and angle sensors, it is important to explore the properties of ferromagnetic thin films grown on nonrigid deformable substrates. In this study, representative deformable substrates (polyimide (PI), polyethylene naphthalate (PEN), and polydimethylsiloxane (PDMS)), which can be bent or stretched, were used to assess the in-plane magnetic field angle-dependent properties of amorphous Ta/CoFeB/MgO/Ta thin films grown on deformable substrates. The effects of substrate roughness, tensile stress, deformable substrate characteristics, and sputtering on magnetic properties, such as the coercive field (*H*_c_), remanence over saturation magnetization (*M*_r_/*M*_s_), and biaxial characteristics, were investigated. This study presents an unconventional foundation for exploring deformable magnetic sensors capable of detecting magnetic field angles.

## 1. Introduction

Because of their characteristics, such as anisotropic magnetoresistance (AMR) [1,2,3,4], giant magnetoresistance (GMR) [5,6,7], tunnel magnetoresistance (TMR) [8,9], Hall effect [10,11], and magnetoimpedance [12,13], magnetic thin films and sensors have garnered significant attention. Currently, studies have focused on analyzing and improving the magnetic properties of magnetic thin films [14,15]. These properties are affected by the spins, orbits, and lattices of the thin films, as well as their interactions [16]. Additionally, the dominant effect of the interfaces of the thin films influences their properties. Studies have used these properties to assess the magnetic anisotropy characteristics such as perpendicular magnetic anisotropy (PMA) [17,18,19], which normally needs to be very thin thickness, and in-plane magnetic anisotropy (IMA), which occurs in relatively thick films [16,20]. Currently, several research groups aim to understand the alterations in these properties to fulfill a long-term goal of implementing these changes along with the basic magnetic properties of thin films in various applications.

Magnetic sensors are used in numerous applications, such as automobiles, home appliances, mobile devices, machines, and medical devices. Additionally, they are particularly useful for sensing changes in motion. In general, magnetic sensors are used in conjunction with magnets to detect changes in magnetic fields. For example, covering the circumference of a rotating object with magnets allows the magnetic sensor to detect changes in the magnetic field based on its rotation. Magnetic sensors can also be used without magnets; however, their method of application depends on their detectable magnetic field range or sensitivity. A geomagnetic sensor that is used as a compass must detect extremely weak geomagnetic fields. Therefore, magnetic sensors are required for a wide range of applications, which depend on their manufacturing methods. Developing new fabrication techniques can increase their applicability.

Recently, bendable and flexible materials have attracted increasing attention because of their potential applicability in wearable electronics and sensors [21], soft robots [22], light-emitting devices [23], and flexible solar cells [24]. Mechanical strain sensing can be applied to conventional spintronic devices [6,25,26,27]. Additionally, TMR [28,29,30] and AMR [1,3] sensors can be fabricated on deformable substrates. The stress of deformation caused by the bending and stretching of the substrates can be transferred to ferromagnetic thin films, inducing compressive or tensile strain. Several studies have demonstrated that magnetic anisotropy can be tuned by assessing the strain on magnetic thin films [31,32,33]. Recently, it has been demonstrated that growing magnetic thin films on deformable substrates can alter their magnetic anisotropy [6,27,34]. However, numerous limitations emerge when annealing or patterning magnetic thin films on deformable substrates due to the precautions that are implemented when using high temperatures and chemical processes. To manufacture a flexible device, a deformable material is used as a substrate. The flexibility of the substrate is an essential property for selecting the method of deformation and the materials to be formed on the substrate. This is because a thin film can be removed if the material on which it has been grown has poor deformability or poor adhesion to the substrate. The process of magnetic thin film formation on a deformable substrate is important; however, the method used to analyze the alterations in properties when bending or stretching is applied to a device is also important.

To study a deformable magnetic sensor (or a deformable in-plane angle sensor in which the rotation axis is in the normal direction of the sample plane), the magnetic properties of a magnetic thin film on a deformable substrate were analyzed. Amorphous Ta/CoFeB/MgO/Ta thin films, which are widely used as ferromagnetic thin films [11,30,35], were sputtered on numerous flexible and stretchable substrates such as polyethylene naphthalate (PEN), polyimide (PI), and polydimethylsiloxane (PDMS). Amorphous Ta/CoFeB/MgO/Ta thin films grown on a deformable substrate were used as blanket films without annealing or patterning. Ta was used as a seed layer to improve the adhesion between the substrate and Ta/CoFeB/MgO/Ta thin film. A sputtering process capable of physically forming a strong thin film was then applied. The magnetic properties of biaxial *H*_c_ and *M*_r_*/M*_s_ were assessed based on the characteristics of the substrate and the phase difference when the PDMS sample was stretched. Simultaneously, analyses of amorphous Ta/CoFeB/MgO/Ta thin films were performed to propose methods for their application in various magnetic sensors or in-plane magnetic field angle sensors that are flexible or stretchable.

## 2. Materials and Methods

### 2.1. Sample Fabrication

One rigid (Si/SiO_2_) and three deformable substrates (PI, PEN, and PDMS), each with an area of 4 mm × 4 mm and thickness of approximately 0.6 mm, were used (Figure 1). Sputter is a Korea Vacuum Tech (Gimpo, Republic of Korea) product (Figure 2). To investigate the magnetic properties of the amorphous CoFeB thin films sputtered onto the deformable substrates, magnetic thin films of Ta(4)/CoFeB(7)/MgO(2)/Ta(2) (thickness, nm) were grown on the substrates. A 4 nm-thick Ta was used as the seed layer and MgO(2)/Ta(2) was used as the cap layer. CoFeB (purity of 99.99%; Kojundo Chemical Laboratory Co., Ltd., Saitama, Japan) was 10 cm in diameter, and 2 mm thick; its elemental ratio was Co:Fe:B = 40:40:20 (Co_40_Fe_40_B_20_). All sputtering targets are products of Kojundo Chemical Laboratory Co., Ltd., Saitama, Japan. Three sputtering guns, tilted at 45° (*θ* = 45°; Figure 2) and mounted 120° (*φ* = 120°) apart, were used. An edge of each substrate was placed parallel to the tangent of the circular sample holder such that the outer direction of the sample holder was 0° (*φ* = 0°) and the inner direction was 180° (*φ* = 180°) to the normal direction of the tangent. The samples were fixed using a sputtering-only metal mask, and a thin film with an area of 3 mm × 3 mm was formed on the substrate using square-shaped holes. Because of this metal mask, no deflection of the deformable substrate occurred during or after sputtering. The diameter of the sample holder was 15 cm.

Following the pre-sputtering process, the Ta/CoFeB/MgO/Ta thin films were grown on the substrates via DC-RF magnetron sputtering at a base pressure of 3 × 10^−6^ Torr. This was achieved by placing the substrate on a sample holder, which was rotating at 10 rpm (Figure 1a and Figure 2). During sputtering in high-purity Ar gas (99.99 %), the working pressure, power, and gas flow rate differed depending on the material. The working pressure, power, and gas flow rate for Ta were 10 mTorr, 20 W, and 10 sccm, respectively. Regarding CoFeB and MgO, the corresponding working pressures (4 and 0.8 mTorr), powers (50 and 150 W), and gas flow rates (10 and 4 sccm) were applied.

### 2.2. PDMS Substrate Fabrication

The sole substrate necessitating fabrication was the PDMS substrate, which had the potential to be torn. PDMS (Dow Corning, Midland, MI, USA) was prepared at a base-to-curing agent volume ratio of 10:1; the mixture was degassed for 1 h. A 2 cm × 2 cm glass substrate was cleaned with acetone, methanol, and deionized water for 15 min under ultrasonication. Following cleaning, a positive photoresist (PR) AZ GXR-601 (Merck, Darmstadt, Hesse, Germany) was spin-coated at 1500 rpm for 30 s, then baked on a hotplate at 95 °C for 1 min. The degassed PDMS mixture was then spun at 500 rpm for 20 s on PR-coated glass and cured on a hotplate at 90 °C for 1 h. To cure the PDMS film completely, it was kept undisturbed for 24 h. Finally, acetone was used to remove the PR and detach the cured PDMS film from the glass. The overall fabrication process is shown in Figure 1c.

### 2.3. Measurement

To identify the crystal structures of the Ta/CoFeB/MgO/Ta thin films sputtered on each substrate, X-ray diffractometry (XRD; SmartLab, Rigaku, TX, USA) was performed on substrates with and without Ta/CoFeB/MgO/Ta thin films. XRD was performed using a Cu target source with a wavelength of 0.15412 nm and an X-ray power of 9 kW (45 kV, 200 mA). The surface profiles and roughness of the substrates were measured using atomic force microscopy (AFM; XE100, PSIA, Suwon, Republic of Korea). The magnetic properties, including the saturation magnetization (*M*_s_) and coercive field (*H*_c_), were measured using superconducting quantum interference vibrating sample magnetometry (SQUID-VSM; MPMS3 system provided by KBSI, Quantum Design, CA, USA). A magnetic hysteresis loop (M-H loop), which was based on the angle, was determined using SQUID-VSM by applying a magnetic field from 0° (*φ* = 0°) to 360° (*φ* = 360°) in the in-plane direction of the sample (Figure 1b). Magnetic force microscopy (MFM; XE100, PSIA, Suwon, Republic of Korea) was used to investigate the microscopic magnetic properties, such as the magnetic phase of the magnetic domains in the thin films, of the stretchable PDMS substrates with tensile strain. The tip of the MFM probe was coated with cobalt and magnetized in the perpendicular direction.

## 3. Results and Discussion

### 3.1. Optical Analysis via XRD

The amorphous properties of Ta/CoFeB/MgO/Ta thin films sputtered on deformable substrates were characterized using XRD (Figure 3). Because all Ta/CoFeB/MgO/Ta thin films were sputtered without annealing, the CoFeB thin film was assumed to have an amorphous structure. Figure 3a,b illustrate the XRD peaks of the substrates and those of the Ta/CoFeB/MgO/Ta thin films grown on the substrates, respectively. Figure 3c shows detailed XRD data of the PDMS substrate and the Ta/CoFeB/MgO/Ta thin film sputtered on the PDMS substrate. Similarly, Figure 3d shows the XRD data of the PI substrate and the Ta/CoFeB/MgO/Ta thin film sputtered on the PI substrate. The XRD data of all samples containing Ta/CoFeB/MgO/Ta thin films sputtered on substrates were similar; additionally, the peaks of these samples were the same as those of the substrates alone. Regarding the PEN substrate, peaks were observed at approximately 27° and 56° [36]. Peaks corresponding to the PDMS and PI substrates were observed at 11° [37] and 21° [38], respectively. Other peaks, such as those of Ta, CoFeB, and MgO, were not observed, except for those of the substrates. The amorphous thin films exhibited broad and low peaks in the XRD graphs [39]; therefore, they were concealed by the peaks of the higher-intensity substrates. Consequently, all peaks in Figure 3 are likely attributed to the substrates. Therefore, the Ta/CoFeB/MgO/Ta thin film had an amorphous structure, regardless of the substrate.

### 3.2. Magnetic Properties with Deformable Substrates

The magnetic properties of the samples that were fabricated by sputtering Ta/CoFeB/MgO/Ta thin films on substrates were assessed to determine the effect of the substrate. The magnetic properties were extracted from the M-H loop, which was determined when an in-plane magnetic field was applied using the SQUID-VSM. All Ta/CoFeB/MgO/Ta thin films that were sputtered onto the substrates exhibited in-plane magnetic anisotropy with rectangular M-H loops in the in-plane field (Figure 4); however, this was not observed in the out-of-plane field (Figure 4) [16]. Excluding the silicon substrates, the squareness of the in-plane M-H loop was poor for samples containing deformable substrates. In particular, the magnetization of the PI sample decreased as the magnetic field was reduced to zero. This behavior was highly correlated to the roughness of the substrate; however, it was also affected by other factors. A detailed discussion and other supporting data are provided in the following sections. The insets in Figure 4 show the low-scale in-plane M-H loops of each sample. The Si/SiO_2_ sample had an in-plane M-H loop that was centered with respect to the magnetic field without an exchange bias, whereas the other samples were off-center. The exchange bias was likely applied by the deformable substrate and sputtering process.

Various magnetic properties, such as the saturation magnetization, remanence, coercive field, magnetic anisotropy field (*H*_k_), and magnetic anisotropy constant (*K*), can be derived from the M-H loop [40]. The magnetic anisotropy constant *K* is an important element in a sensor that detects the magnetic field angle and is an indicator for evaluating magnetic anisotropy. However, the exact hard axis, *M*_s_, and *H*_k_ must be known, and various analyses must be performed to determine the value of *K* [41]. In the formula used to calculate magnetic anisotropy energy (*E*_a_) (Equation (1)), *K*_0_ is an isotropic term that is not affected by the angle, *K*_u_ is the uniaxial anisotropy constant, and *K*_c_ is a magnetocrystalline anisotropy constant that contributes to the four-fold symmetry (or biaxial characteristics) [16,41,42]:(1)$$E_{a}=K_{0}+K_{u}\sin^{2}\left(\theta \right)-\frac{1}{4}K_{c}\sin^{2}\left(2\theta \right)$$

In magnetic thin films with in-plane magnetic anisotropy, a hard axis is formed in the normal direction of most planes via shape anisotropy. This occurs because of their thickness and area [16,40]. Therefore, analyzing the magnetic anisotropy of the normal direction of the plane is not appropriate for assessing sensors that sense the in-plane magnetic field angle. Rather, the magnetic anisotropy field between the in-plane and out-of-plane magnetic anisotropies was obtained for each sample. The magnetic anisotropy energy was not calculated separately. The magnetic anisotropy fields were approximately 8.2 kOe (Si/SiO_2_), 6.3 kOe (PDMS), 3.5 kOe (PI), and 13 kOe (PEN).

The crystallization of the Ta/CoFeB/MgO/Ta thin films did not proceed through annealing. Magnetic sensors, especially in-plane magnetic field angle sensors, were the focus of this study. Therefore, only the angular dependence of *H*_c_ and *M*_r_*/M*_s_, which can be identified based on the substrate, was analyzed. In general, variations in *H*_c_ indicate that the pinning effects of the magnetic domain movements are present or absent owing to substrate roughness or impurities [40]. Additionally, variations in *M*_r_*/M*_s_ imply that multi-grains or magnetic domains are composed of magnetic anisotropies of various orientations [43,44]. These values (*H*_c_ and *M*_r_*/M*_s_) can be used to qualitatively evaluate the magnetic anisotropy of a sample [38,45,46].

Magnetizations (*M*) were normalized to the saturation magnetization (*M*_s_~1060 emu/cc) of each sample with the substrate (Figure 4 and Figure 5a,c). This was carried out to compare the variation in magnetization across different magnetic field angles of each sample. Therefore, *M/M_s_* was used as the *y*-axis of the M-H loop. In this study, the M/M_s_-H loop has been simplified to the M-H loop for readability. Samples that were applied magnetic fields at 0° and 90° are shown in Figure 5a,c, respectively. Excluding the Si/SiO_2_ substrate that was sputtered with the Ta/CoFeB/MgO/Ta thin film, a magnetic field applied at 0° (Figure 5a) did not result in significant differences in the shape of the M-H loop between substrates. For the Si/SiO_2_ sample, the rectangular loop shape was the most prominent among the four samples (Si/SiO_2_, PDMS, PI, and PEN). The rectangular shape of the M-H loop indicates that the sample has good magnetic anisotropy and rapid magnetic switching properties. However, the shapes of the M-H loops differed between substrates at 90° (Figure 5b). Additionally, the *H*_c_ and *M*_r_*/M*_s_ values differed between samples (Table 1). *H*_c_ increased as the angle of the magnetic field increased from 0° to 90°. The largest variation was observed in the PEN sample; *H*_c_ increased by approximately 4-fold from *H*_c,0°_ = 20.96 Oe to *H*_c,90°_ = 84.35 Oe. This value (*H*_c,90°_ = 84.35 Oe) was approximately 6.6 times higher than the 90° value of the Si/SiO_2_ sample (*H*_c,90°_ = 12.65 Oe) (Table 1). This can be explained by the roughness of the substrate surface. The pinning effect intensified as the roughness increased; eventually, *H*_c_ increased in proportion with the increase in roughness. The Si/SiO_2_ and PDMS samples with comparable roughness demonstrated similar *H*_c_ values; the PI and PEN samples with large roughness also had large *H*_c_ values (Figure 5c,d). The range of the *y*-axis scale at a magnetic field of 0° was smaller than that of the 90° magnetic field (Figure 5b,d). Curve fitting shows that both angles exhibited similar tendencies.

The effect of magnetic field angle variation on *H*_c_ is difficult to explain using only the pinning effect. Generally, *M*_r_*/M*_s_ values are affected by the varying magnetic anisotropy orientations of multi-grains, whereas the *H*_c_ values are affected by their grain size [47]. Therefore, the effect was noticeable for deformable substrates, which have rough and uneven surfaces, because it is easy for these surfaces to form multi-grains from magnetic thin films by physically dividing areas. Compared with the Si/SiO_2_ and PDMS samples, which have smooth surfaces, the PI and PEN samples had *H*_c_ values that varied between the magnetic field angles of 0° and 90°. This variation was also observed at angles that were not between 0° and 90°. Similarly, *M*_r_*/M*_s_ values at 90° were lower than those at 0° because the magnetic anisotropy of each grain was formed in many directions. *M*_r_*/M*_s_ and *H*_c_ varied continuously from magnetic field angles of 0°to 360°. The biaxial characteristics were likely influenced by the multi-grains.

### 3.3. Angular Dependence of H_c_ and M_r_/M_s_

Because this study focused on using magnetic sensors as in-plane angle sensors, the M-H loops of all samples were measured in the in-plane directions at 30° intervals from 0° to 360° (see Figure 6 and Figure 7, Table 2).

The in-plane easy axes of all samples were formed via shape anisotropy because of the thin thickness of the amorphous CoFeB thin film (7 nm). Therefore, a hard axis was formed perpendicular to the plane; thus, the Si/SiO_2_ sample showed isotropic properties for both *H*_c_ and *M*_r_*/M*_s_. However, its shape was not completely isotropic in the polar coordinate system. This effect was attributed to the sputtering system. The sputter was equipped with a 45° (*φ* = 45°, Figure 2) tilted sputter gun, and the sample holder rotated at 10 rpm. Therefore, the energy applied to the substrate during sputtering differed based on the direction of the sample. Consequently, the stress transferred to the thin film differed depending on the direction of the sample [48].

This stress and strain are related to the magnetostriction effect. In particular, CoFeB exhibited a large magnetostriction constant (*λ*_s_ = 31 × 10^−6^) [40,49]. Therefore, the sample sputtered on the Si/SiO_2_ substrate did not fully exhibit isotropic properties. In the PDMS substrate sample, which had a marginally rougher surface than that of the Si/SiO_2_ sample, *M*_r_*/M*_s_ was somewhat isotropic; however, *H_c_* was not. In contrast, the PI and PEN samples exhibited uniaxial or biaxial characteristics. *H*_c_ of the PI sample exhibited maximum values at 90° and 270° (Figure 6c). Minimum values were observed at 0° and 180° for the uniaxial properties. The *M*_r_*/M*_s_ values of the PI samples were high at 0°, 120°, 180°, and 300°. Low *M*_r_*/M*_s_ values were observed at 60°, 150°, 240°, and 330°; therefore, it exhibited a four-fold symmetry (or biaxial characteristics). The PEN sample exhibited biaxial characteristics for *H*_c_ and *M*_r_*/M*_s_ (Figure 6d). The *H*_c_ of the PEN sample was mostly uniaxial because of the difference between the values at 90° (or 270°) and 0° (or 180°), whereas the *M*_r_*/M*_s_ values were similar at 0°, 90°, 180°, and 270°. *H*_c_ and *M*_r_*/M*_s_ exhibited low values at 60°, 120°, 240°, and 300°. The PEN sample exhibited complete biaxial characteristics for both *H*_c_ and *M*_r_*/M*_s_. The detailed M-H loops of the PI and PEN samples, based on the applied magnetic field angles, are shown in Figure 7a,b, respectively. For visibility, only angles ranging from 0 to 150° were used.

The biaxial characteristics of the PI and PEN samples were also analyzed owing to the influence of the sputtering method. Because the samples were not annealed, it was difficult to observe the effect of magnetocrystalline anisotropy. However, they were not patterned to exhibit shape-anisotropy characteristics. The large *H*_c_ and *M*_r_/*M*_s_ values at 90° and 270° in the PEN sample were not explained by the high roughness of the substrate. Compared with the rigid Si/SiO_2_ substrates, PDMS, PI, and PEN were deformable substrates characterized by their soft and uneven surfaces. These substrates exhibited a strong interaction with thin films when they were strained. Therefore, the influence of the stress between the substrates (PDMS, PI, and PEN) and the thin film was stronger than that of the influence observed in the Si/SiO_2_ sample. Additionally, each substrate has different characteristics. However, the Ta/CoFeB/MgO/Ta thin film might have been formed in a directional shape to enable shape anisotropy on the deformable substrate because of the stress and non-rigid surface of the deformable substrates. Therefore, it was concluded that the Ta/CoFeB/MgO/Ta thin film may have biaxial anisotropy because of shape anisotropy; this was due to the directional shape. The shape might have been formed at the micrometer level or lower.

If the causes of these phenomena can be clearly identified and applied, deformable magnetic sensors can be used in various applications. The magnetoresistance depends on the magnetic anisotropy and is related to *H*_c_ and *M*_r_/*M*_s_ [29,50,51,52]. For example, controlling magnetocrystalline anisotropy through annealing or forming shape anisotropy by patterning a magnetic thin film will enable the fabrication of effective magnetic sensors or in-plane angle sensors.

### 3.4. Strain Effects in the CoFeB Thin Film Sputtered on the PDMS Substrate

The effect of the physical deformation (tensile strain in one direction) of the Ta/CoFeB/MgO/Ta thin film sputtered on the PDMS substrate was investigated at the macroscopic and microscopic scales using SQUID-VSM and MFM, respectively. For each measurement, all stretched samples were fixed on 4 mm × 4 mm glass substrates at 5% strain using double-sided tape. This method was required to accurately measure the stretchable samples. However, the magnitude of the detected magnetic moment during SQUID-VSM analysis may differ owing to the variation in the position of the Ta/CoFeB/MgO/Ta thin film on the sample holder; therefore, a normalization operation is recommended. In addition, the sample must be firmly attached to the glass substrate to prevent separation due to the rapid vibration (80 Hz) that is transmitted to it during SQUID-VSM analysis. Hence, measuring the variations in the properties by bending the sample was challenging; for these reasons, it was not performed. Moreover, the distance between the sample and the tip of the MFM probe must be constant during MFM analysis; however, maintaining the bent sample at a constant distance was also challenging.

The overall in-plane M–H loops are shown in Figure 8a. The *H*_c_ values of the PDMS samples before and after strain application were not significantly different. However, the *H*_c_ of the sample marginally increased with tensile strain, suggesting that a few grains formed anisotropy in the direction opposite to that applied by the magnetic field. The strain affected the magnetic anisotropy orientation because of the induced tensile stress [53]. This was confirmed by the histogram of the with-strain condition, which was high and narrow around the 0° phase in the out-of-plane direction (Figure 8b). The strength of the out-of-plane magnetization characteristics of the PDMS sample increased with increasing strain. The PDMS sample with good squareness of the in-plane M-H loop, whose magnetization decreases near *M*_r_, began exhibiting some out-of-plane characteristics. This was indicated by the reduction in the *M*_r_/*M*_s_ values (Figure 8a and Table 3). This behavior was widely distributed based on the phase in Figure 8b,c. The magnetic phase only deviated from 0° under with-strain conditions; the phase shifted by 0.04°. Therefore, the tensile or surface stress of the substrate can alter the direction of the magnetic anisotropy; this was proven by the reduction in *M*_r_ of the in-plane M-H loop.

The histogram of the after-strain samples was wider than those of the with- and before-strain histograms. The orientation of the magnetic anisotropy was likely affected by the increasing unevenness of the surface of the magnetic thin film. This occurred because of the fine scratches that occurred after stretching the sample. Therefore, it is recommended that the sample is artificially damaged, until the magnetic anisotropy orientation is stabilized, before it is used as a mechanical strain sensor. This can be accomplished by repeating the tensile operation several times. More precise measurements are possible if a mechanical strain sensor based on spintronic devices is used in conjunction with the existing mechanical strain sensors.

The *H*_c_, *M*_r_*/M*_s_, and magnetic phases were tuned by applying tensile stress to the magnetic thin film, indicating that the magnetic anisotropy orientation can be tuned by strain. Controlling magnetic anisotropy is essential for adjusting the performance of magnetic or mechanical strain sensors [1,3].

## 4. Conclusions

The magnetic properties of ferromagnetic materials sputtered onto various deformable substrates must be studied for application in deformable or wearable magnetic sensors. CoFeB is a suitable ferromagnetic layer material for magnetic sensors because of its large magnetic moment and tunable magnetic anisotropy. The magnetic properties of amorphous CoFeB thin films sputtered on various substrates varied, even when the same amorphous CoFeB thin film was used. These properties can be exploited in various magnetic sensor applications.

The roughness of the substrate has a significant impact on *H*_c_ because of the pinning effect of the magnetic domains. The *H*_c_ and *M*_r_/*M*_s_ values of the samples depend on the magnetic field angle. The *H*_c_ of the PEN sample at a magnetic field angle of 90° was 6.6 times higher than that of the Si/SiO_2_ sample when the magnetic field angle was 0°. Although the CoFeB thin film was not annealed and did not exhibit magnetic anisotropy, the biaxial properties of *H*_c_ and *M*_r_/*M*_s_ were affected by the stress between the deformable substrates and the Ta/CoFeB/MgO/Ta thin films; this was influenced by sputtering. The biaxial properties of *H*_c_ and *M*_r_/*M*_s_ are important because they can qualitatively explain that magnetic anisotropy energy is able to have the biaxial properties. In particular, the *M*_r_/*M*_s_ values of the PI and PEN samples exhibited biaxial characteristics. Additionally, the magnetic phase shifted by 0.04° at a tensile strain of 5%. Upon the relaxation of the strain, the magnetic anisotropy of the grains did not return to its original state; however, the average magnetic phase did not vary.

These effects can be appropriately applied to magnetic or deformable angle sensors.

## Figures and Tables

**Figure 1 sensors-23-07479-f001:**
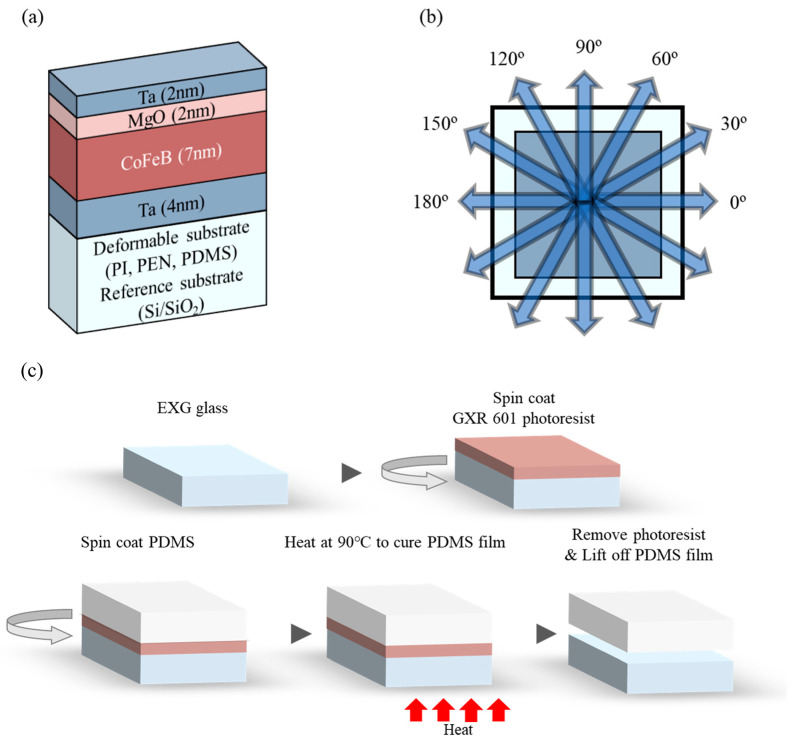
Schematic images: (**a**) Sample stack on four substrates; (**b**) in-plane magnetic field angle about the sample (top view); (**c**) PDMS substrate fabrication.

**Figure 2 sensors-23-07479-f002:**
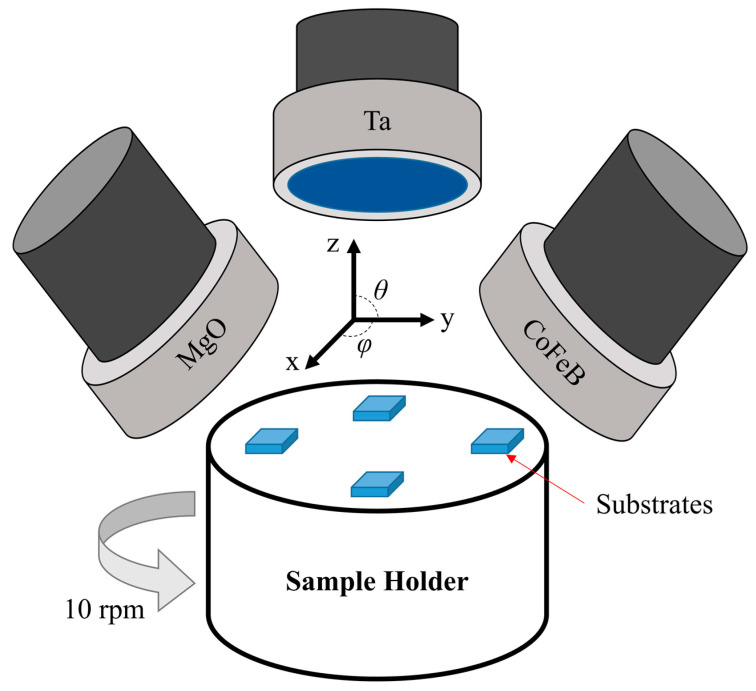
Schematic of a sputtering equipment comprising three sputter guns, targets, a sample holder, and substrates.

**Figure 3 sensors-23-07479-f003:**
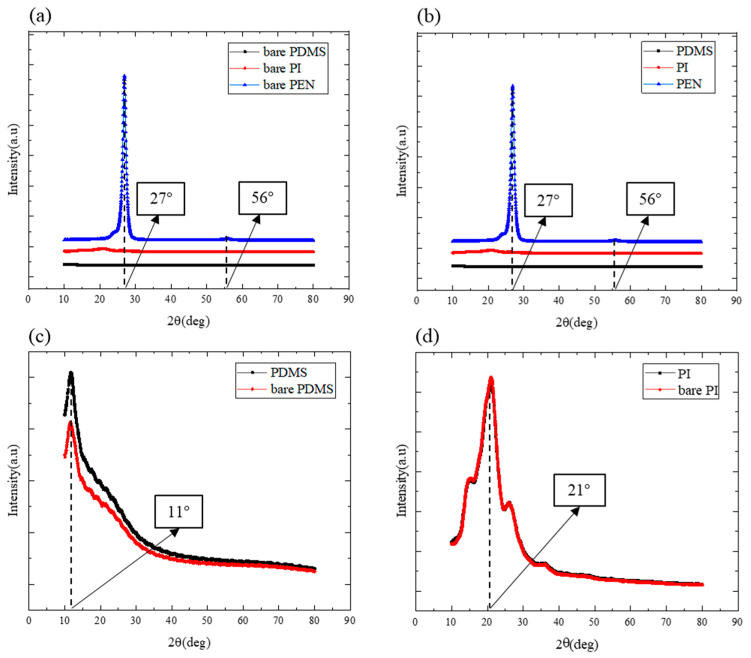
X-ray diffraction (XRD) data for (**a**) Si/SiO_2_, PDMS, PI, and PEN substrates; (**b**) Si/SiO_2_, PDMS, PI, and PEN samples sputtered with the Ta/CoFeB/MgO/Ta thin films; (**c**) PDMS substrates and the sputtered PDMS samples; and (**d**) PI substrates and the sputtered PI samples.

**Figure 4 sensors-23-07479-f004:**
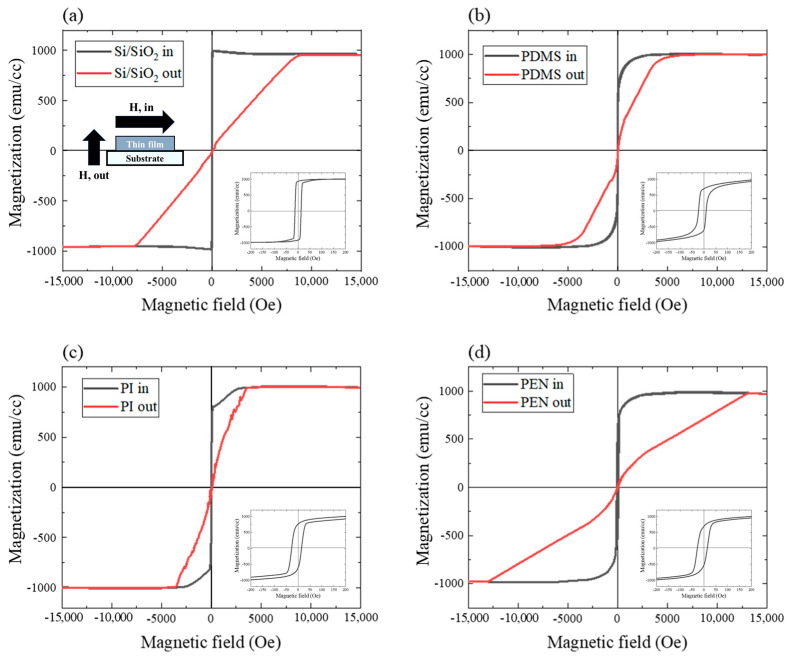
M-H loops with in-plane magnetic fields (black line), out-of-plane magnetic fields (red line), and low-scale M-H loops (inset) for (**a**) Si/SiO_2_, (**b**) PDMS, (**c**) PI, and (**d**) PEN.

**Figure 5 sensors-23-07479-f005:**
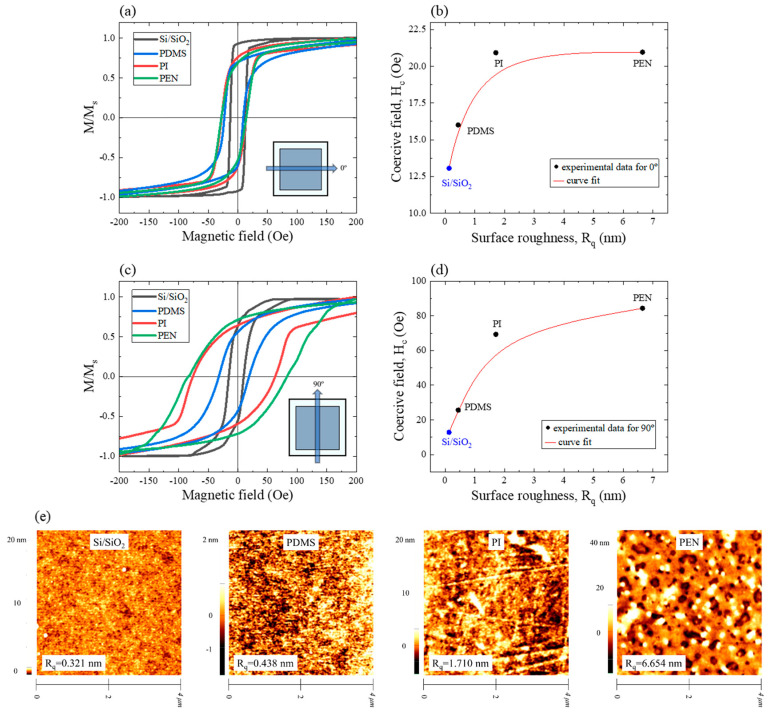
M-H loops of the samples on in-plane magnetic fields of (**a**) 0° and (**c**) 90°; *H*_c_ of the samples vs. surface roughness (*R*_q_) of the substrates for (**b**) 0° and (**d**) 90°; (**e**) *R*_q_ of the substrates—Si/SiO_2_, PDMS, PI, and PEN.

**Figure 6 sensors-23-07479-f006:**
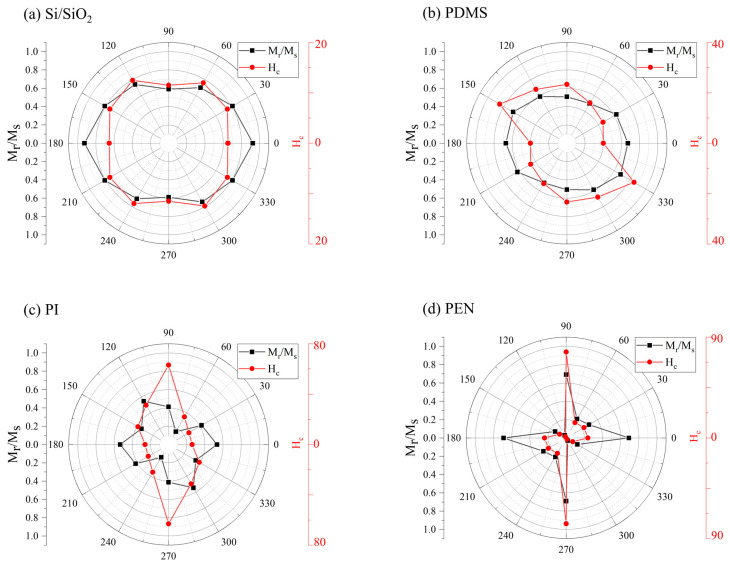
Angle dependence of *H*_c_ and *M*_r_/*M*_s_ from 0° to 360° in a polar coordinate system: (**a**) Si/SiO_2_; (**b**) PDMS; (**c**) PI; and (**d**) PEN.

**Figure 7 sensors-23-07479-f007:**
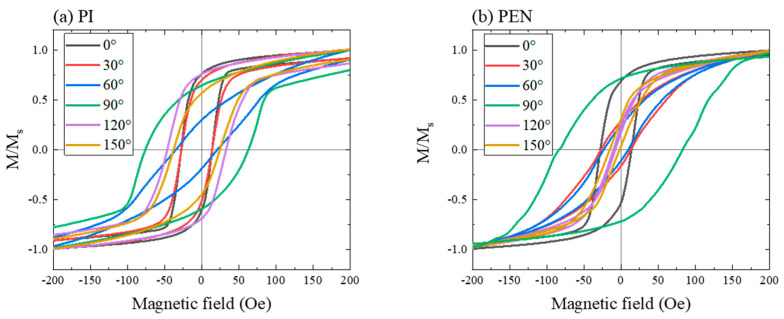
M-H loops with magnetic field angles from 0° to 150° for the (**a**) PI and (**b**) PEN samples.

**Figure 8 sensors-23-07479-f008:**
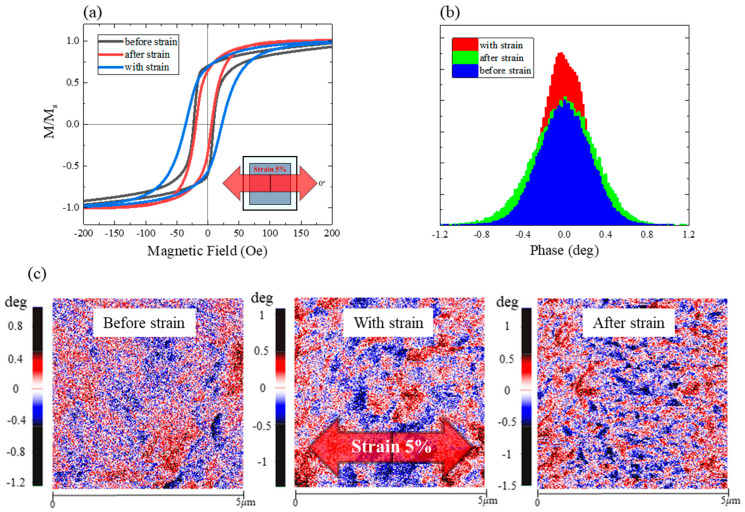
(**a**) M–H loops, (**b**) histograms of the MFM phase, and (**c**) MFM phase maps of the PDMS samples measured before, with, and after applying tensile strain (5%).

**Table 1 sensors-23-07479-t001:** *R*_q_, *H*_c_, and *M*_r_*/M*_s_ values of the substrates at 0° and 90°.

Substrate	*R*_q_*,* nm	*H*_c_, Oe		*M* _r_ */M* _s_	
		0°	90°	0°	90°
Si/SiO_2_	0.13	13.06 ± 0.16	12.65	0.93	0.59
PDMS	0.44	16.01 ± 7.24	25.66	0.70	0.03
PI	1.71	20.92 ± 6.88	69.30	0.76	0.18
PEN	6.65	20.96 ± 6.95	84.35	0.69	0.62

**Table 2 sensors-23-07479-t002:** *H*_c_, and *M*_r_*/M*_s_ of the substrates from 0° to 150°.

Angle	*H*_c_, Oe		*M* _r_ */M* _s_	
	PI	PEN	PI	PEN
0°	0.259	0.239	0.533	0.691
30°	0.258	0.227	0.420	0.290
60°	0.349	0.194	0.162	0.243
90°	0.866	0.937	0.413	0.690
120°	0.496	0.016	0.546	0.034
150°	0.389	0.083	0.345	0.141

**Table 3 sensors-23-07479-t003:** The *H*_c_ and *M*_r_/*M*_s_ values of the PDMS sample before, with, and after strain application.

Condition	*H* _c_	*M*_r_/*M*_s_
Before strain	16.12	0.645
With strain	29.07	0.625
After strain	12.41	0.485

## Data Availability

Not applicable.

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
