# Peer review of "Magnetic Properties of Amorphous Ta/CoFeB/MgO/Ta Thin Films on Deformable Substrates with Magnetic Field Angle and Tensile Strain"

_sensors, 2023, doi:10.3390/s23177479_

Round 1

Reviewer 1 Report

The current work illustrates the magnetic properties of CoFeB thin films. Microstructure investigations have been done by using MFM combined with the study of in-plane magnetic anisotropy. I found the manuscript very interesting and will make a good contribution to the scientific community. Thus, I recommend accepting the manuscript to be published in Sensors, after the authors maintain these necessary points.

The introduction:

The introduction needs to be enhanced with relevant references which have an improving the perpendicular magnetic anisotropy in pure ferromagnetic materials and their alloys with different physical forms. The authors can use these suggested references to enhance the introduction in the revised version of the manuscript:

1-     Li, H.; Xie, Y.; Yang, H.; Hu, H.; Li, M.; Li, R.-W. The Effect of Size and Strain on Micro Stripe Magnetic Domain Structure of CoFeB Thin Films. Metals 202313, 678. https://doi.org/10.3390/met13040678

2-     Salaheldeen, M.; Wederni, A.; Ipatov, M.; Zhukova, V.; Lopez Anton, R.; Zhukov, A. Enhancing the Squareness and Bi-Phase Magnetic Switching of Co2FeSi Microwires for Sensing Application. Sensors 202323, 5109. https://doi.org/10.3390/s23115109

 3-     Silva, A.S., Sá, S.P., Bunyaev, S.A. et al. Dynamical behaviour of ultrathin [CoFeB (tCoFeB)/Pd] films with perpendicular magnetic anisotropy. Sci Rep 11, 43 (2021). https://doi.org/10.1038/s41598-020-79632-0.

 4-     Salaheldeen, M.; Goyeneche, M.L.M.; Alvarez-Alonso, P.; Fernandez, A. Enhancement the perpendicular magnetic anisotropy of nanopatterned hard/soft bilayer magnetic antidot arrays for spintronic application. Nanotechnology 202031, 485708.

 5-     Salaheldeen, M.; Garcia-Gomez, A.; Corte-Leon, P.; Ipatov, M.; Zhukova, V.; Gonzalez, J.; Zhukov, A. Anomalous Magnetic Behavior in Half-Metallic Heusler Co2FeSi Alloy Glass-Coated Microwires with High Curie Temperature. J. Alloys Compd. 2022923, 166379.

 6-      https://doi.org/10.1063/5.0106414

 Materials and methods:

1-      The authors should add more information and details in the experimental part and how they estimate the layer thickness of the film. If possible, could add the commercials sources of the materials and targets used in the current work? In addition, should the author describe more details about the sputtering condition and if they applied any magnetic field during the deposition process?

2-      Figure 2 it is better to move to the materials and method.

3-     For PDMS Substrate Fabrication, the authors said, “The degassed PDMS mixture was then spun at 500 rpm for 20 sec on PR-coated glass and cured on a hotplate at 90 °C for 1 hour.” Could please add explanations why they used 90 °C for 1 hour? Why not higher or lower this temperature and time?

4-     The authors mentioned in the manuscript that they used VSM for M-H loops, thus it is necessary to plot the M-H loops in all the manuscripts with (emu/cm3) to see the real value of the magnetization instead of the normalized magnetizations.

 Results:

1-      The authors should add the uncertainties for all outcome data in the tables and the figures to see the real behavior of all parameters studied in the current manuscript.

2-      For more accurate values could the authors plot the M—H loops with the term od emu/CC instead of normalized loops?

3-      For M-H loops plotted in Figure 4, it will be great if the authors plot a low scale of M-H and insert it as an inset in Figure 4 as the loops with the current form do not indicate any information about the coercivity and Mr.

4-      The authors did not explain the distortions that appeared in the in-plane M-H loop in Figure 4c. An explanation is needed in the revised version of the manuscript.

5-      The authors should add a table summarizing all information that can extract from in/out-of-plane loops, such as Ms, Hk, Mr, Keff, Ku, and Hc. They only present one or two parameters; the rest of the parameters are needed in the revised version of the manuscript.

6-      The in-plane M-H loops in Figure 4a,c need to represent in a better way. As the black loops (Si/SiO2) appear uncentred and some exchange bias fields are found, please declare these points.

7-       MFM analysis needs more discussion and explanations. The main purpose of MFM analysis declares the OOP configurations, the samples show strong in-plane magnetic anisotropy with not found OOP component, could the authors explain in detail the origin of the out-of-plane signal in the MFM images?

8-       The authors should add the grid lines in Figure 7 and Figure 8.  

Author Response

Thanks to your comments. We revised the manuscript and figures, and wrote a response letter for them. The response letter contains everything for all editors and reviewers to see at once.

Yours sincerely,
Byeong-Kwon Ju

Reviewer 2 Report

The authors must check again whole article for preventing mistakes.

Author Response

(The authors gave the same response as above.)

Round 2

Reviewer 1 Report

The authors fix most my comments and I see the manuscript is suitable for publish in the present form.

Reviewer 2 Report

The article has been suitably revised.